# Effect of Callus Cell Immobilization on the Textural and Rheological Properties, Loading, and Releasing of Grape Seed Extract from Pectin Hydrogels

**DOI:** 10.3390/gels10040273

**Published:** 2024-04-17

**Authors:** Elena Günter, Oxana Popeyko, Fedor Vityazev, Sergey Popov

**Affiliations:** Institute of Physiology of Federal Research Centre, Komi Science Centre, Urals Branch of the Russian Academy of Sciences, 50, Pervomaiskaya Str., 167982 Syktyvkar, Russia; opopeyko@mail.ru (O.P.); rodefex@mail.ru (F.V.)

**Keywords:** pectin, callus, cells, hydrogel, texture, rheology, delivery system, grape seed extract

## Abstract

The purpose of the present study was to prepare pectin hydrogels with immobilized *Lemna minor* callus cells and to identify the effect of cell immobilization on the textural, rheological, and swelling properties; loading; and releasing of grape seed extract (GSE) from the hydrogels. Hardness, adhesiveness, elasticity, the strength of linkage, and complex viscosity decreased with increasing cell content in the hydrogels based on pectin with a degree of methyl esterification (DM) of 5.7% (TVC) and during incubation in gastrointestinal fluids. An increase in the rheological properties and fragility of pectin/callus hydrogels based on pectin with a DM of 33.0% (CP) was observed at a cell content of 0.4 g/mL. TVC-based pectin/callus beads increased their swelling in gastrointestinal fluids as cell content increased. TVC-based beads released GSE very slowly into simulated gastric and intestinal fluids, indicating controlled release. The GSE release rate in colonic fluid decreased with increasing cell content, which was associated with the accumulation of GSE in cells. CP-based beads released GSE completely in the intestinal fluid due to weak textural characteristics and rapid degradation within 10 min. Pectin/callus hydrogels have the ability to preserve GSE for a long time and may have great potential for the development of proanthocyanidin delivery systems due to their novel beneficial physicochemical and textural properties.

## 1. Introduction

Pectin hydrogels are widely used in the food industry to create food hydrogels, achieve a certain food texture, stabilize food emulsions, act as fat and starch substitutes, and deliver bioactive compounds [1,2,3,4]. Pectin is stable when passing through the upper gastrointestinal tract and is destroyed by enzymes of the bacterial microflora of the colon. Therefore, pectin hydrogels can serve as promising carriers of grape seed extract (GSE) proanthocyanidins in the large intestine.

GSE proanthocyanidins have the ability to scavenge free radicals in the human body and are therefore considered powerful antioxidants [5,6,7]. GSE has anti-obesity, anti-diabetic, anti-cancer, and anti-inflammatory activities [8,9,10,11,12]. GSE is used as an antioxidant dietary supplement [13]. GSE has the potential to improve jejunal health by suppressing inflammation and regulating alkaline phosphatase [14], increasing goblet cell numbers and reducing myeloperoxidase levels in the colon [15], improving the gut microbiota [12,15], and it may also protect against oxidative stress by slowing oxidation reactions in the intestine [16]. The use of proanthocyanidins in the food industry is difficult due to their features, such as their low stability in biological fluids, poor water solubility, and easy exposure to light intensity, temperature, relative humidity, and pH [6,12]. To improve the stability and bioavailability of proanthocyanidins during transit through the gastrointestinal tract, an encapsulation method was used [17]. Various systems have been used to encapsulate grape seed proanthocyanidins, including polylactic acid nanoparticles [18], alginate/cellulose microcapsules [19], alginate/pectin aerogel microspheres [20], *Bletilla striata* polysaccharide/chitosan microspheres [21], pectin/pullulan films [22], alginate/chitosan microparticles [23,24], chitosan/lecithin microspheres [25], and chitosan particles [26]. Developed encapsulating agents have facilitated the controlled release of polyphenols with varying degrees of success. The applied carriers often demonstrate the rapid release of polyphenols within the first 60 min, not providing sustained release at the target site [20,27,28]. In this regard, the creation of improved biodegradable carriers of proanthocyanidins is necessary to ensure their targeted delivery to the intestine and maintain high antioxidant activity.

In this study, we intended to obtain GSE-loaded hydrogels based on pectins that differ in structure, into which callus cells were to be immobilized. The inclusion of callus culture cells in the hydrogels and the use of pectins with different structures will make it possible to obtain hydrogels with a given texture that is different from that of hydrogels based on pectin alone. Moreover, the resulting hydrogels may serve as a delivery system for the antioxidant dietary supplement GSE. The developed hydrogels could be useful for creating functional foods with antioxidant properties based on hydrocolloids with the desired texture and rheological properties.

To provide plant-based food to the ever-growing population of the planet, it is important to develop new technologies for the production of healthy and diverse foods of plant origin [29]. In recent years, plant cell cultures have been seen as a new approach to producing plant foods [29,30,31,32,33,34,35]. Cell immobilization, which allows cells to adapt to environmental conditions, is a valuable tool to improve cell productivity [30]. One of the strategies for plant cell immobilization is encapsulation in hydrogels. Callus cells and isolated plant cells are used for the 3D bioprinting of food hydrogels based on alginate, carrageenan, pectin, and agar [30,31,32,33,34,35]. A few works have been carried out to prepare various plant cell-containing hydrogels to create diverse food textures and produce innovative plant-based food products [29,30,31,32,33,34,35]. Callus cells with turgor pressure incorporated into a hydrogel can have the unique texture of artificial plant tissues [31,32]. The question of how the immobilization of callus cells in a hydrogel affects the textural and rheological characteristics of the latter remains poorly understood. We previously showed that Ca/alginate hydrogels with immobilized callus cells reduced swelling and GSE release in gastrointestinal environments due to the less porous hydrogel structure and the GSE retention in the cells [36]. In the present study, duckweed *Lemna minor* L. callus cells were used as model cells immobilized in pectin hydrogels because these cells have previously shown properties that allow GSE to be encapsulated in alginate/callus hydrogels and delivered to the colon.

The purpose of the present study was to prepare pectin hydrogels with immobilized *Lemna minor* callus cells and to identify the effect of cell immobilization on the textural, rheological, and swelling properties; loading; and releasing of GSE from the hydrogels.

## 2. Results and Discussion

### 2.1. Characterization of GSE-Loaded Hydrogel Beads

Pectin hydrogel beads with immobilized plant cells were obtained from pectin (2.0%), duckweed callus cells (LM) (0.1–0.4 g/mL), and GSE (1.0 mg/mL) using the cross-linking agent calcium lactate (1.0%). Pectins with varying degrees of methyl esterification were used as sources of pectins, in particular tansy callus pectin (TVC) with a DM of 5.7% and commercial citrus pectin (CP) with a DM of 33.0%. The addition of callus cells to the hydrogel resulted in a slight change in the diameter of the beads (4.10–4.68 mm) (Table 1). At the same time, the cell content in the TVC-based beads was positively correlated with the diameter of the beads (R^2^ = 0.979, *p* < 0.05). A similar pattern was previously established for alginate particles with immobilized callus cells, which showed an increase in the alginate particle diameter in proportion to the cell content in them [36]. This correlation was not observed in the case of the CP pectin-based beads (R^2^ = −0.502, *p* < 0.05). The hydrogels contained conglomerates of living callus cells, which was confirmed by determining cell viability, which was 95%.

The stability of hydrogel beads was studied during incubation in distilled water at 4 °C for 5 days. After one day of incubation, the weight of hydrogel beads 2.0TVC (control), 0.1LM-2.0TVC, 0.2LM-2.0TVC, 0.3LM-2.0TVC, and 0.4LM-2.0TVC increased by 21, 40, 59, 52, and 21%, respectively, compared to the initial weight (taken as 100%). After five days of incubation, the weight increases were 25, 69, 80, 69, and 31%, respectively. The hydrogel beads showed stability because they did not collapse within 5 days. The smallest weight increase was observed for 2.0TVC and 0.4LM-2.0TVC. Hydrogel beads 2.0CP (control), 0.1LM-2.0CP, 0.2LM-2.0CP, 0.3LM-2.0CP, and 0.4LM-2.0CP increased in weight by 11, 32, 28, 53, and 28%, respectively, after one day of incubation compared to the initial weight (taken as 100%). After five days of incubation, the weight of the hydrogels practically did not change because the increases were 14, 35, 30, 53, and 30%, respectively. Thus, the hydrogel beads based on CP pectin swelled intensively during the first day and then remained stable and did not degrade for 5 days. In addition, the CP-based hydrogel beads exhibited less weight gain in water compared to those of the TVC-based ones.

Scanning electron micrographs of freeze-dried pectin/Ca (2.0TVC and 2.0CP) and pectin/callus hydrogel beads loaded with GSE are presented in Figure 1. The pectin/Ca beads without callus cells (used as controls) had an elongated shape (Figure 1a,f). At the same time, the pectin/callus beads had a shape that was close to spherical, with the exception of 0.1LM-2.0CP, which had an elongated shape (Figure 1g). The pectin/Ca beads without cells and pectin/callus beads with a low cell content (0.1 g/mL) were characterized by a surface morphology with long folds and wrinkles (Figure 1a,b,f,g). With the increase in the cell contents of the hydrogel beads, the surface morphology became a cellular structure (Figure 1c,d,h–j) or rough and bumpy (Figure 1e). The formation of folds, wrinkles, bumps, and a cellular structure on the surface of the beads could be due to the presence of GSE in them. It was previously shown that smooth polysaccharide/chitosan microspheres and pectin/Zn/alginate gel particles were transformed into wrinkled microspheres after loading with proanthocyanidins [21,37].

Irrespective of the cell content, the phenolic compound contents in the hydrogel beads based on TVC pectin were within a close range (38.32–43.16 μg/mL) (Table 1). Simultaneously, the pectin/callus hydrogel beads based on CP pectin had a 1.4-fold higher phenolic compound content than the 2.0CP beads without cells. There was a noticeable rise in the amount of phenolic compounds in the beads based on CP pectin as the cell concentration increased. A positive correlation was revealed between these parameters (R^2^ = 0.860, *p* < 0.05). This could be explained by GSE getting inside the callus cells when they were loaded into pectin/callus hydrogel beads and the callus cells’ subsequent ability to retain the GSE. We previously reported this phenomenon for alginate hydrogels with immobilized callus cells [36].

There was minimal variation in the encapsulation effectiveness (EE) of GSE in the hydrogel beads based on TVC (Table 1). A little rise in EE was seen as the ratio of TVC pectin to cells increased, most likely as a result of the high encapsulating ability of low methylesterified TVC pectin. Additionally, it has been demonstrated that in gel formulations, the EE of GSE increases as the alginate concentration rises [36]. Compared to the cell-free 2.0CP beads, the EE of the GSE in the pectin/callus beads based on CP pectin was 1.3 times higher. This was probably caused by the callus cells’ ability to retain GSE. Moreover, it is likely that the inclusion of the callus cells promoted the maintenance of GSE by preventing the formation of pores in the pectin hydrogels [36]. It was shown that the EE depended on the structure of pectin; in particular, cell-free beads from TVC pectin with a lower DM (5.7%) and a higher Mw (666 kDa) had a higher EE (75.1%) compared to those based on CP pectin (DM 33%, Mw 402 kDa, and EE 57.7%). The incorporation of callus cells into the CP-based hydrogels led to a 25–29% rise in EE because the cells accumulated GSE. Proanthocyanidin EE was previously shown to be 85% in gelatin/pectin microparticles [38], 50–89% in *Bletilla striata* polysaccharide/chitosan microspheres [21], 10–16% in alginate/pectin microspheres [39], and 95% in pectin/Zn/alginate gel particles [37].

### 2.2. Rheological Properties of GSE-Loaded Pectin/Ca and Pectin/Callus Hydrogels

Hydrogel samples with a minimum (0.1 g/mL) and maximum (0.4 g/mL) content of callus cells, as well as hydrogels without cells (controls), were selected to study the rheological properties. The storage modulus G′ of all TVC- and CP-based hydrogels was greater than the loss modulus G″ in the entire LVE region, and their values depended on the pectin used and the cell content in the hydrogels (Figure 2a,b). The G′_LVE_ value of the 2.0TVC hydrogel was higher than that of the 2.0CP hydrogel, indicating that the gel network strength of the TVC-based hydrogel was higher. The addition of 0.1 g/mL cells to the TVC-based hydrogel caused the G′_LVE_ and G″_LVE_ to increase by 1.4–1.5 times, and an increase in the cell content to 0.4 g/mL caused a decrease in these parameters by 1.5–1.9 times (Figure 2a). This indicates that increasing the cell content in TVC-based hydrogels resulted in a decrease in the gel network strength. This could be due to the loosening of the hydrogel structure due to the immobilization of cells in the hydrogel and the interference in the convergence of pectin chains to form a three-dimensional hydrogel network [35,36]. The addition of 0.1 g/mL cells to the CP-based hydrogels had no effect on G′_LVE_ and G″_LVE_ (Figure 2b). At the same time, a higher cell content (0.4 g/mL) caused an increase in these parameters by 3.0–4.4 times, which indicated an increase in the gel network strength.

The cell wall surface is negatively charged due to the predominance of free carboxyl groups of galacturonic acid residues present in the primary cell wall and middle lamella [40]. The cell wall has the properties of an ion exchanger. Due to the predominance of negative fixed charges in the cell wall, the cations are concentrated. Due to cation exchange properties, the cell wall serves as a site for the accumulation of cations, which can interact with the COO^−^ groups of TVC or CP pectins. This could lead to the formation of a stronger gel network in the 0.1LM-2.0TVC and 0.4LM-2.0CP samples. In addition, the pectins present in the cell wall of duckweed callus cells interacted with Ca^2+^ ions during the preparation of the pectin/callus hydrogels, which also led to the strengthening of the hydrogels.

Storage modulus (G′) and loss modulus (G″) values, obtained depending on the frequency in the LVE region, are presented in Figure 2c,d. All the G′ and G″ values of the TVC-based hydrogels were higher than those of the CP-based hydrogels, indicating that the gel network strength of the TVC-based hydrogels was higher. The addition of 0.1 g/mL cells to the TVC or CP hydrogels did not affect the G′ and G″ values. At the same time, the addition of 0.4 g/mL cells to the TVC- and CP-based hydrogels led to a decrease and increase in these parameters, respectively. Similar changes in the values of G′ and G″ depending on the cell concentration were observed in k-carrageenan gels enriched with thawed lupine callus tissue [35].

The loss tangent Tan [δ]_LVE_ for all TVC- and CP-based hydrogels was 0.26–0.35 and 0.10–0.21, respectively, indicating that they exhibited solid-like behavior. The loss tangent slope in the nonlinear region (Tan[δ]_AF_) showed that the samples had transitioned from the gel-like to the liquid state. The ratio of the viscous and elastic components for the studied samples at high strain values showed similar Tan[δ]_AF_ values, which indicated similar hydrogel spreadability (Table 2). The values of k″/k′ in the frequency range under study for all samples were below unity (0.13–0.32), and low values of the moduli n′ and n″ (0.10–0.18 and 0.03–0.96, respectively) indicated the elastic nature of the resulting hydrogels (Table 2). All the TVC-based hydrogels showed lower γFr values (0.42–0.65) compared to the CP-based hydrogels (1.19–5.62), which could indicate that the TVC-based hydrogels have greater fragility. Moreover, adding callus cells to the 2.0TVC hydrogel did not affect its fragility, whereas increasing the cell content in the 2.0CP hydrogel led to an increase in its fragility.

The rheological characteristics of hydrogels, representing properties such as the strength of linkage, the number of linkages, the timescale of the junction zone, and the distance of linkage, were described according to Alghooneh et al. [41] (Table 2). The G′_LVE_, corresponding complex modulus (G*_FP_); limiting value of stress (τL); and the frequency dependences of the elastic (k′), loss (k″), and complex (A) moduli in the 2.0TVC hydrogel were higher than those in the 2.0CP hydrogel, indicating an increase in the strength of linkage in the TVC hydrogel (Table 2).

The rheological parameters, such as the ratio of maximum complex modulus to linear complex modulus (G*max/G*_LVE_), the fracture stress (τFr), and the frequency dependences of the elastic (n′) and complex (z) moduli, represent the number of linkages [41]. The τFr value was 4.9 times higher in the 2.0TVC hydrogel than in the 2.0CP hydrogel, indicating a greater number of linkages in the 2.0TVC hydrogel. The G*max/G*_LVE_, n′, and z parameters did not differ in the 2.0TVC and 2.0CP hydrogels.

The timescale of the junction zone was expressed by rheological parameters such as Tan [δ]_LVE_, the overall loss tangent (k″/k′), the limiting value of strain (γL), the fracture strain (γFr), and the slope of complex viscosity (η*s) [41]. Tan [δ]_LVE_, η*s, and k″/k′ increased in the TVC hydrogel, which could indicate an increase in the timescale of the junction zone due to the larger number of linkages in it.

The distance of linkage parameter, expressed by the frequency dependences of the loss moduli (n″), was 13.7 times lower in the 2.0TVC hydrogel than in the 2.0CP hydrogel.

The addition of 0.1 g/mL cells to the TVC hydrogel led to an increase in the strength of linkage, as the corresponding parameters (G′_LVE_, G*_FP_, τL, k′, k″, and A) increased (Table 2). In addition, the number of linkages in these pectin/callus hydrogels increased, since the τFr parameter was higher, although other parameters (G*max/G*_LVE_, z, and n′) were close. Rheological characteristics such as the timescale of the junction zone and the distance of linkage did not change. The addition of 0.1 g/mL cells to the CP hydrogel did not cause changes in rheological parameters, with the exception of a decrease in the distance of linkage (n″). Moreover, almost all of the indicated rheological parameters were higher in the 0.1LM-2.0TVC hydrogel than in the 0.1LM-2.0CP hydrogel, which indicates that the former hydrogel has a higher gel network strength.

The addition of a high concentration of cells (0.4 g/mL) to the TVC hydrogel resulted in a decrease in the strength of linkage due to a decrease in G′_LVE_, G*FP, τL, k′, k″, and A. The number of linkages, the timescale of the junction zone, and the distance of linkage did not change in the 0.4LM-2.0TVC hydrogel. The addition of 0.4 g/mL cells to the CP hydrogel resulted in an increase in the strength of linkage due to an increase in G′_LVE_, G*FP, τL, k′, k″, and A. The values of G*max/G*LVE, n′, and z were close, but the τFr parameter increased in the 0.4LM-2.0CP hydrogel, which could indicate an increase in the number of linkages. The timescale of the junction zone (Tan [δ]_LVE_, k″/k′, and η*s) increased and the distance of linkage (n″) decreased in the 0.4LM-2.0CP hydrogel.

Complex viscosity as a function of frequency for GSE-loaded pectin hydrogels based on callus cells and TVC or CP pectins is presented in Figure 3. Complex viscosity was higher for all the TVC-based hydrogel samples compared to the CP-based hydrogels. The addition of 0.1 g/mL cells to hydrogels based on TVC or CP did not affect their values. At the same time, the addition of 0.4 g/mL cells to hydrogels based on TVC and CP resulted in a decrease and increase in complex viscosity, respectively.

Thus, the incorporation of callus cells into the pectin hydrogels influenced their rheological characteristics, such as the strength of linkage, the number of linkages, the timescale of the junction zone, the distance of linkage, and complex viscosity.

### 2.3. Texture Analysis of GSE-Loaded Hydrogels

The initial hardness of the GSE-loaded pectin-Ca gel beads based on TVC pectin (2.0TVC) was 4.7 times higher than that of those based on CP pectin (2.0CP) (Figure 4a and Figure 5a). This was likely due to the lower DM of TVC pectin, which led to the presence of a large number of non-methoxylated carboxyl groups in pectin macromolecules, which bind calcium ions, and the formation of a stronger gel [37,42,43]. In addition, the high hardness of the TVC pectin gels could be due to the higher MW and the amount of galactose and arabinose residues. It was previously shown that gels formed from high Mw pectin have high gel strength [42]. The formation of a stronger hydrogel could be due to an increase in hydrogen bonds in pectin as a result of a greater number of arabinose and galactose residues included in the pectin side chains. The initial adhesiveness, elasticity, and work of the 2.0TVC hydrogel were higher than those of the 2.0CP hydrogel by 1.4, 1.4, and 4.8 times, respectively (Figure 4b–d and Figure 5b–d).

The texture analysis data regarding the hardness of hydrogels are consistent with the rheological analysis data. An increase in the strength and number of linkages, the timescale of the junction zone, complex viscosity, and a decrease in the distance of linkage in the 2.0TVC hydrogel were observed compared to the 2.0CP hydrogel (Table 2).

The incorporation of living callus cells into hydrogels based on TVC or CP pectin caused a decrease in the initial hardness, work, adhesiveness, and elasticity (Figure 4 and Figure 5) due to the loosening of the hydrogel structure when callus cells were embedded into the hydrogel and the convergence of pectin chains in the hydrogels being interfered with. We have previously shown a similar tendency for changes in the gel strength of alginate hydrogels with immobilized callus cells of campion and duckweed [36]. In addition, a decrease in the hardness of k-carrageenan and agar gels enriched with thawed lupine callus tissue has been shown [35,44]. Carrot callus cells decreased the gel strength of the agar gel [32]. Increasing the concentration of encapsulated living lettuce leaf cells in low methoxylated pectin gel reduced the mechanical force [31].

The decrease in the elasticity of hydrogels could be associated with an increase in their fragility. Our rheological analysis also showed a 1.9–4.8-fold increase in the fragility of the CP-based hydrogels with increasing cell content. At the same time, the fragility of the TVC-based hydrogels was higher than that of the CP hydrogels by 2.8–8.6 times and did not depend on the cell content.

A more significant decrease in the values of the textural parameters (hardness, work, adhesiveness, and elasticity) occurred in the pectin/callus hydrogels derived from TVC compared to those derived from CP, which was probably due to the fact that the CP-based hydrogels were initially weaker. At the same time, these textural characteristics decreased with an increase in the cell content. A negative correlation was shown between the cell content in the TVC-based hydrogel and the hardness (R^2^ = −0.999, *p* < 0.05), work (R^2^ = −0.981, *p* < 0.05), adhesiveness (R^2^ = −0.878, *p* < 0.05), and elasticity (R^2^ = −0.965, *p* < 0.05) of the hydrogels. The texture data are supported by the rheological analysis data, which showed a decrease in the strength of linkage and complex viscosity when adding a high concentration of cells (0.4 g/mL) to the TVC hydrogel, indicating a reduction in the rheological properties of the hydrogel.

The hardness of the hydrogels based on weaker CP pectin decreased slightly (by 8–17% only) with the addition of 0.1–0.4 g/mL cells. For the CP hydrogels, no correlation was established between the cell content in the hydrogels and the textural characteristics of the hydrogels, with the exception of adhesiveness (R^2^ = −0.868, *p* < 0.050). At the same time, an increase in the timescale of the junction zone, the strength of linkage, the number of linkages, and complex viscosity, as well as a decrease in the distance of linkage in the CP hydrogel, was observed with the addition of 0.4 g/mL of cells, which was probably due to the interaction of the cations on the cell surface with the COO^−^ groups of pectin and the formation of a stronger gel network.

The hardness, work, elasticity, and adhesiveness of all hydrogels decreased after incubation in the simulated gastric (SGF, pH 1.25, 2 h), intestinal (SIF, pH 7.0, 4 h), and colonic (SCF, pH 6.8 + pectinase, 1 h) fluids (Figure 4 and Figure 5). This indicates that the hydrogels’ textural characteristics deteriorated during incubation in the gastrointestinal fluids. This may be due to the presence of Na^+^ ions in gastrointestinal fluids, which replace Ca^2+^ ions in the pectin gel network [19]. As a result, a decrease in cross-linking in the hydrogels, partial destruction of the gel network, and a change in the textural characteristics of the hydrogels occurred. Moreover, the pectinase included in SCF destroyed α-1,4-bonds between galacturonic acid residues, which led to the destruction of the pectin gel network and, as a consequence, the deterioration of the textural characteristics. The hardness of the TVC-based hydrogels decreased in SGF, SIF, and SCF by 1.5–3.4, 4.0–9.6, and 9.3–204.1 times, respectively, compared to the initial hardness (Figure 4a). The work of TVC-based hydrogels decreased in SGF, SIF, and SCF by 1.4–4.1, 4.0–10.5, and 13.5–920.0 times, respectively (Figure 4b). The elasticity of TVC-based hydrogels decreased in SGF, SIF, and SCF by 1.1–1.5, 1.1–1.6, and 1.6–1.9 times, respectively (Figure 4d). The adhesiveness of the 2.0TVC hydrogel without cells and pectin/callus hydrogels with a low cell content (0.1LM-2.0TVC) decreased in SGF, SIF, and SCF by 1.4–1.6, 1.2–1.3, and 1.1–1.3 times, respectively, compared with their initial adhesiveness (Figure 4c). At the same time, the adhesiveness of pectin/callus hydrogels with high cell contents (0.3LM-2.0TVC and 0.4LM-2.0TVC) increased in SGF, SIF, and SCF by 1.1, 1.4, and 1.6 times, respectively, compared to the initial adhesiveness. The hardness, work, and elasticity of the CP-based hydrogels decreased after incubation in SGF by 3.8–11.4, 4.5–19.4, and 1.3–2.7 times, respectively. At the same time, the adhesiveness increased by 1.1–1.4 times in SGF in comparison to the initial value.

### 2.4. Swelling of GSE-Loaded Pectin/Callus and Ca/Pectin Gel Beads

The swelling behavior of wet GSE-loaded gel beads based on callus cells and TVC or CP pectins in SGF, SIF, and SCF is presented in Figure 6. The TVC- and CP-based hydrogel beads did not swell in the acidic environment of SGF, which was attributed to the shrinkage of the hydrogels under acidic conditions [24,45]. In an acidic environment, due to an excess of H^+^ ions, the protonation of pectin carboxyl groups (COO^−^) occurred, which caused a decrease in electrostatic repulsion, the penetration of liquid into the hydrogel, and swelling [27,45].

In SIF, the TVC-based hydrogel beads gradually swelled in the first 3 h and then began to degrade (Figure 6a). When the pH of the medium increased (7.0), the COO^−^ groups were deprotonated, which caused an increase in electrostatic repulsion and the expansion of the pectin gel network [46,47]. Greater swelling in polyphenol-loaded pectin/alginate microspheres [27], repaglinide-loaded pectin/alginate beads [45] and Zn/alginate gel particles [48] was also found at a high pH (6.0–6.8).

When callus cells were added at a low concentration (0.1 g/mL) to the pectin hydrogel beads, the swelling behavior in SIF was similar to that of the 2.0TVC beads without cells (used as a control) (Figure 6a). At the same time, the addition of higher concentrations of callus cells (0.2–0.4 g/mL) to the hydrogel led to an increase in the swelling of the pectin/callus beads in comparison with the 2.0TVC beads (Figure 6a). This phenomenon is probably associated with the significant decrease in the textural (hardness, work, and elasticity) and initial rheological (the strength of linkage and complex viscosity) characteristics of the pectin/callus hydrogels with increasing cell content in the hydrogels (Figure 4). As a result, the hydrogel network became weaker, and the liquid diffused into the hydrogel beads. In addition, the hardness, work, and elasticity of all hydrogels decreased after incubation in SIF. At the same time, the adhesiveness of the pectin/callus hydrogels with higher cell contents (0.2LM-2.0TVC, 0.3LM-2.0TVC, and 0.4LM-2.0TVC) increased in SIF compared to the initial adhesiveness, which could have increased the ability of the hydrogels to be wetted by liquid and swell (Figure 4c). An increase in the degree of bead swelling occurred with an increase in the cell content in the hydrogels from 0.1 to 0.3 g/mL. Increasing the cell content in the hydrogels to 0.4 g/mL caused a decrease in bead swelling compared to that at 0.3 g/mL. This was due to the low pectin-to-cell ratio, which resulted in slower swelling of the pectin gel and, subsequently, the faster destruction of the beads.

The CP-based hydrogel beads were destroyed within 10 min in SIF (Figure 6b). This was due to the higher DM of CP pectin, which led to fewer non-methoxylated COO^−^ groups, less binding of Ca^2+^ ions, and the formation of a weaker hydrogel, which was quickly destroyed in SIF [42,43]. Moreover, the low gel strength of the CP hydrogels was attributed to the lower MW of pectin. In addition, the rapid degradation of the hydrogel beads was attributed to a significant decrease in textural properties such as hardness, work, and elasticity, as well as an increase in adhesiveness, after incubation in SGF, which may have increased the diffusion of liquid into the hydrogel beads and led to rapid swelling (Figure 5).

In SCF, the TVC-based hydrogel beads degraded at different rates (Figure 6a). In particular, the control cell-free beads (2.0TVC) and beads with a maximum cell content (0.4LM-2.0TVC) were destroyed within 10 min in SCF. At the same time, beads with a cell content of 0.1–0.3 g/mL dissolved gradually during incubation in SCF. Thus, the presence of cells in the hydrogels in an amount of 0.1–0.3 g/mL led to a slower destruction of the hydrogel beads. The rapid degradation of the pectin hydrogel beads was due to the action of pectinase on the α-1,4-bonds in pectin and the breakdown of the gel network. In addition, rapid bead degradation was associated with a significant decrease in the hardness and elasticity of the hydrogels after incubation in SCF.

### 2.5. Fourier Transform Infrared Spectroscopy (FTIR) of GSE-Loaded Hydrogels

The FTIR spectra of grape seed extract (GSE) (a), the GSE-loaded Ca/pectin gel beads (2.0TVC and 2.0CP (controls)) (b, e), and the pectin/callus gel beads (0.1LM-2.0TVC, 0.4LM-2.0TVC, 0.1LM-2.0CP, and 0.4LM-2.0CP) (c, d, f, and g) are presented in Figure 7. In the FTIR spectrum of GSE, the bands at 3417 and 2930 cm^−1^ correspond to the -OH vibrations, which belong to the phenolic structures of GSE and C–H, respectively [19,39,46,49] (Figure 7a). The peaks found at 1612–1110 cm^−1^ correspond to the functional groups of proanthocyanidin polyflavonoids [39,46]. The peaks at 1612, 1518, 1445, and 1286 cm^−1^ are due to the stretching of the aromatic ring [46,49]. The absorption peak at 1351 cm^−1^ is associated with –C–OH vibrations [49]. The peak at 821 cm^−1^ is attributed to phenoxy substitution [49].

In the FTIR spectra of the GSE-loaded Ca/pectin gel beads 2.0TVC (Figure 7b) and 2.0CP (Figure 7e), the bands at 3431 and 3423 cm^−1^ are attributed to the stretching of the –OH groups in the 2.0TVC and 2.0CP gels, respectively [50,51,52]. The peaks at 2924 and 2934 cm^−1^ are attributed to –CH_3_ or C–H vibrations [51,52,53,54]. The peaks at ca. 1740 correspond to the ester carbonyl (C=O) groups [51,52,53,54]. The bands at 1617 and 1612 cm^−1^ indicate the vibration of the COO- and –CH_3_ groups [52,53,54]. The stronger intensity of the carboxyl band at 1617 cm^−1^ compared to the ester carbonyl band at 1743 cm^−1^ indicates a low degree of esterification of TVC pectin [53,54]. The intensity of the peak at 1738 cm^−1^ for the 2.0CP hydrogel was stronger than that at 1743 cm^−1^ for the 2.0TVC hydrogel, which indicates a higher DM of 2.0CP pectin. The peaks in the 1426–1239 and 1421–1237 cm^−1^ regions are attributed to the asymmetric vibrations of –C–O–C– bonds and –CH groups or –CH_3_ groups [52]. The 1096–1017 and 1101–1018 cm^−1^ regions are attributed to the asymmetric –C–O–C– and –C–C bonds [51,52,53,54]. The peak at 960 cm^−1^ is attributed to the symmetric –C–O–C– and –C–C vibrations [52].

The FTIR spectra of the pectin/callus hydrogels revealed a shift in the O–H stretching band by ca. 3400 cm^−1^ towards short waves, which indicates a restructuring of hydrogen bonds within the polymer network (Figure 7c,d,f,g) [23,27]. A decrease in the signal intensity of the FTIR spectra was shown for the pectin/callus hydrogels with high cell contents (0.4LM-2.0CP) (Figure 7g).

The peaks of GSE were mainly overlapped by those of the pectin hydrogels, indicating that GSE was incorporated into the Ca/pectin and pectin/callus hydrogels. A similar observation was previously made by researchers for proanthocyanidin-loading systems such as polysaccharide/chitosan, pectin/Zn/alginate, pectin/pullulan, and alginate/cellulose [19,21,22,37]. The O–H stretching band shift at ca. 3400 cm^−1^ in the GSE-loaded pectin gel spectra compared to the spectra of GSE may be due to the incorporation of GSE into the pectin hydrogels through hydrogen bonding (Figure 7) [37].

### 2.6. The Release of GSE from the Hydrogel Beads

Data regarding the cumulative release of GSE from wet hydrogel beads based on callus cells and TVC or CP pectins in the simulated gastrointestinal environment are presented in Figure 8. All TVC-based hydrogel beads released GSE very slowly in SGF at pH 1.25 (4.7–7.9%) and SIF at pH 7.0 (7.5–10.5%) (Figure 8a). The control cell-free beads (2.0TVC) and beads with maximum cell content (0.4LM-2.0TVC) released GSE in the first hour of exposure to SCF (pH 6.8 + pectinase) due to the breakdown of hydrogels (Figure 8a). Beads containing 0.1–0.3 g/mL cells released GSE gradually during exposure to SCF. Moreover, the 0.1LM-2.0TVC beads released GSE faster than the 0.2LM-2.0TVC and 0.3LM-2.0TVC beads. A negative correlation was found between the callus cell content in the hydrogel beads and the release of GSE in the SCF (R^2^ = −0.792, *p* < 0.05). This indicated that a decrease in the rate of GSE release from hydrogels was associated with an increase in the cell content in them. It is likely that GSE entered callus cells, was retained during loading into the pectin/callus hydrogel beads, and was gradually released in SIF and SCF. A similar trend was previously established for alginate/callus hydrogel particles, which exhibited the retention capacity of callus cells [36].

The rapid GSE release in SCF was likely due to a significant decrease in the hardness and elasticity of the hydrogels, as well as the gradual destruction of hydrogel beads after incubation in SCF under the influence of both medium electrolytes and pectinase (Figure 4a,d). Na^+^ ions from gastrointestinal fluids replaced Ca^2+^ ions in the gel network, resulting in a decrease in cross-link density and the partial destruction of the gel network. At a pH of 6.8, the COO^−^ groups of pectin were deprotonated, and electrostatic repulsion and the expansion of the gel network occurred [48,50]. The liquid diffused faster into the weak hydrogels, and the GSE leaked into the external environment. A similar phenomenon was also shown for alginate/cellulose microcapsules loaded with GSE proanthocyanidins [24], GSE-loaded alginate hydrogel beads with immobilized callus cells [36], and pectin/Zn/alginate gel particles [37]. The faster release of grape seed proanthocyanidins from chitosan particles [26], proanthocyanidins from *Bletilla striata* polysaccharide/chitosan microspheres [21], curcumin from alginate/ZnO beads [48] and chitosan-pectinate nanoparticles [50], and polyphenols from pectin/alginate beads [27] was found at higher environmental pH values (6.0–7.4). In addition, pectinase from SCF destroyed the pectin gel network, which led to the deterioration of the hydrogel textural characteristics and the rapid release of GSE into the environment. It was also previously shown that large amounts of curcumin were released from chitosan/pectinate nanoparticles in pectinase-rich medium at pH 6.4 [50].

Data regarding the cumulative release of GSE from wet hydrogel beads based on callus cells and CP pectin in the simulated gastrointestinal fluids are presented in Figure 8b. All CP-based hydrogel beads released GSE slowly into SGF (8.3–15.9%), while the pectin/callus beads released GSE more slowly than the cell-free CP beads. The CP-based hydrogel beads released GSE faster compared to the TVC-based hydrogel beads, which was likely due to the lower hardness, work, adhesiveness, and elasticity of the CP-based hydrogels. All CP-based hydrogel beads released GSE in the first 10 min of incubation in SIF due to the dissolution of the hydrogel beads. This was due to a significant decrease in some textural properties (hardness, work, and elasticity) after incubation in SGF, the expansion of the gel network as a result of the deprotonation of the COO^−^ groups of pectin at pH 7.0, and the influence of medium electrolytes, which could increase the diffusion of SIF into the beads and their rapid dissolution.

Pectin/callus hydrogel beads based on TVC released GSE into SGF at a significantly lower rate (5–8%) compared to *Bletilla striata* polysaccharide/chitosan microspheres (16–39%) [21], chitosan particles (88%) [26], alginate/callus hydrogels (16–36%) [36], alginate/cellulose microcapsules releasing proanthocyanidins (21%) [19], and alginate/ZnO beads releasing curcumin (17%) [47]. In addition, TVC-based pectin/callus hydrogel beads released GSE into SIF slower (8–11%) than previously obtained *Bletilla striata* polysaccharide/chitosan microspheres (38–54%) [21], chitosan particles (91%) [26], alginate/callus hydrogels (36–49%) [36], alginate/cellulose microcapsules releasing proanthocyanidins (52%) [24], and alginate/ZnO beads releasing curcumin (42%) [48]. The developed pectin/callus hydrogel beads showed the controlled release of GSE in artificial gastrointestinal media of different pHs, meaning that they have superior advantages compared to previously developed proanthocyanidin delivery systems.

## 3. Conclusions

Hydrogel beads based on pectins of different structures and immobilized callus cells were obtained. FTIR spectroscopy and SEM, texture, and rheological analyses were performed to compare the physicochemical characteristics of the hydrogels. FTIR spectroscopy showed that GSE was incorporated into the Ca/pectin and pectin/callus hydrogels. The incorporation of callus cells into pectin hydrogels caused a decrease in the initial hardness, work, adhesiveness, and elasticity of these hydrogels. A more significant decrease in textural parameters occurred in the pectin/callus hydrogels based on TVC (DM 5.7%) compared to the weaker pectin/callus hydrogels based on CP (DM 33.0%). The textural and rheological (the strength of linkage and complex viscosity) characteristics decreased with increasing cell content in the TVC-based hydrogels due to the loosening of the hydrogel structure and the convergence of pectin chains in the hydrogels being interfered with. At the same time, an increase in rheological properties (the strength of linkage, the number of linkages, the timescale of the junction zone, and complex viscosity), an increase in fragility, and a decrease in the distance of linkage in the CP hydrogels were observed with the addition of 0.4 g/mL of cells. The textural characteristics of the hydrogels deteriorated during incubation in gastrointestinal fluids. The TVC- and CP-based hydrogel beads did not swell in the acidic environment of SGF. The addition of higher concentrations of callus cells (0.2–0.4 g/mL) to the TVC-based hydrogels resulted in an increase in the swelling of the hydrogel beads in SIF due to a significant reduction in the textural and rheological characteristics with increasing cell content. The presence of cells in the hydrogel in an amount of 0.1–0.3 g/mL led to a slower destruction of the TVC-based hydrogel beads in SCF. The CP-based hydrogel beads were destroyed within 10 min in SIF due to the formation of a weaker hydrogel. The TVC-based hydrogel beads released GSE very slowly into SGF and SIF, indicating the controlled release of GSE. The GSE release rate in SCF depended on the cell content in the hydrogel beads. A decrease in the GSE release rate in SCF occurred with an increase in cell content from 0.1 to 0.3 g/mL. This could be due to the accumulation of GSE in callus cells and its gradual release during incubation in SCF. The CP-based hydrogel beads released GSE faster compared to the TVC-based hydrogel beads in SGF and completely released GSE in SIF, which was attributed to the weak textural and rheological characteristics of CP hydrogels. The developed pectin/callus hydrogel beads demonstrated the controlled release of GSE at different pH values in simulated gastrointestinal media, so they have great potential for retaining GSE in large quantities and for a long time. Pectin/callus hydrogel systems may have great potential for the development of proanthocyanidin delivery systems due to their novel physicochemical and textural properties.

## 4. Materials and Methods

### 4.1. Materials

TVC pectin from the callus culture of *Tanacetum vulgare* L. was taken from the Institute of Physiology of the Federal Research Center “Komi Science Center of the Urals Branch of the Russian Academy of Sciences”. The chemical characteristics of pectin have been established previously (Table 3) [55]. Citrus pectin LC-S18XH (CP) was obtained from JRS Silvateam Ingredients S.r.l., Bergamo, Italy. The grape seed extract (GSE), containing 7.45% monomers (4.35% catechin and 3.10% epicatechin) and 95% proanthocyanidins, was purchased from Foodchem International Corporation, Shanghai, China. All other chemicals were of analytical grade.

### 4.2. Callus Culture of Lemna minor L.

The callus culture of duckweed *Lemna minor* L. was maintained at 24 °C in the dark for 21 days in a modified Murashige and Skoog’s medium [56] containing sucrose (30 g/L), agar (8 g/L), 6-benzylaminopurine (0.5 mg/L), and 2.4-dichlorophenoxyacetic acid (1.0 mg/L). The callus culture was taken from the collection of callus cultures of the Institute of Physiology of the Federal Research Center “Komi Science Center of the Urals Branch of the Russian Academy of Sciences”.

### 4.3. Preparation and Characterization of Hydrogel Beads

Hydrogel beads based on TVC pectin from callus culture (2.0%) or commercial citrus pectin CP (2.0%), cells of *L. minor* callus culture (LM) (0.1, 0.2, 0.3, 0.4 g/mL), and GSE (1.0 mg/mL) were prepared using calcium lactate (1.0%). Pectin (2.0%) was dissolved in distilled water. Next, 1 mL of GSE dissolved in ethanol (20 mg/mL), the final concentration of which was 1.0 mg/mL, was added to the pectin solution. LM callus cells at concentrations of 0.1, 0.2, 0.3, and 0.4 g/mL were added to this mixture and mixed. The resulting mixture was extruded into a calcium lactate cross-linking solution (1.0%) using a nozzle (diameter 3.0 mm and distance 5 cm). As a result, hydrogel beads were obtained. These beads were then incubated for 4 h at 10 °C and subsequently washed three times with distilled water. Cell-free 2.0TVC and 2.0CP hydrogel beads were used as controls. The composition of the hydrogel beads is presented in Table 1.

An optical microscope (Altami, St. Petersburg, Russia) with a camera was used to determine bead size. For fifteen wet beads of each formulation, the projected equivalent diameter (circle diameter with equivalent area) was measured using an image analysis program (ImageJ 1.46r program, National Institutes of Health, Bethesda, MD, USA).

The viability of callus cells in the hydrogel was determined by the Evans blue exclusion staining method [30]. Briefly, the hydrogels were cut into thin sheets (1 mm thick), and a few drops of Evans blue solution were applied to them for 30 s at 25 °C, which stained dead callus cells. The percentage of living cells was determined using the same optical microscope.

Scanning electron micrographs of freeze-dried gel beads were taken using a scanning electron microscope (SEM) (Tescan Vega3 SBU, Brno, Czech Republic) at 54× magnification (scale bar 1 mm) and 20 kV.

To determine the phenolic compound (PC) contents of the hydrogel beads, 1.0 g of wet beads was combined with 8 mL of distilled water and homogenized for 5 min, and then 2 mL of 96% ethanol was added. Next, the mixture was exposed to ultrasound for 15 min. To 600 μL of the resulting mixture, 1800 μL of distilled water and 150 μL of the Folin–Ciocalteu reagent were added [57]. This mixture was incubated for 5 min at 25 °C. Next, 450 μL of a saturated solution of Na_2_CO_3_ (47 g/100 mL) was added, mixed, and incubated in the dark for 2 h at 25 °C. The optical density of the samples was measured at a wavelength of 765 nm. The PC content was measured using a gallic acid calibration graph (1–100 μg/mL).

To study the stability of the hydrogel beads, they were incubated in distilled water at 4 °C for 5 days. The hydrogel beads (*n* = 6) were weighed on an analytical balance on days 1 and 5. The percentage changes in weight compared to the initial weight (taken as 100%) were determined.

### 4.4. Rheological Properties

LM callus cells at concentrations of 0.1 and 0.4 g/mL were added to a mixture of pectin (2.0%) and GSE (1 mg/mL) and mixed. The resulting mixtures were placed in dialysis bags with a pore size of 14 kDa (Sigma-Aldrich Co, St. Louis, UO, USA) and incubated in a calcium lactate cross-linking solution (1.0%) for 72 h at 10 °C. As a result, cylindrical hydrogels were obtained, which were removed from the dialysis bags, washed, and cut into 1 mm high disks. Cell-free hydrogels 2.0TVC and 2.0CP, prepared in a similar manner, were used as controls.

Rheological measurements, including the determination of strain and frequency sweep, were carried out on a rotational rheometer (Anton Paar, Physica MCR 302, Graz, Austria) equipped with parallel geometry plates (diameter 25 mm; gap 1 and 3.0 mm). Using a controlled shear rate mode at 20 °C with a constant frequency and stress of 1 Hz, a strain sweep evaluation was carried out for strain amplitudes ranging from 0.01 to 100%. The G′_LVE_, G″_LVE_, Tan [δ]_LVE_ in LVE, G*_LVE_, γL, γFr, τL, G*max/G*_LVE_, τFP, G*FP, τFr, and Tan [δ]_AF_ values were determined as described previously [41,58,59,60,61]. The frequency dependences of k′, k″, k″/k′, n′, n″, A, z, and η*_S_ were calculated [62].

### 4.5. Texture Analysis

The textural properties of wet hydrogel beads (hardness, work, adhesiveness, and elasticity) were determined using a Texture Analyzer (TA-XT Plus, Texture Technologies Corp., Stable Micro Systems, Godalming, UK) and Texture Exponent 6.1.4.0 software (Stable Micro Systems, Godalming, UK). The hydrogel beads (*n* = 20) were punctured with a P/2 probe.

### 4.6. Fourier Transform Infrared Spectroscopy (FTIR)

FTIR spectra of pectin/callus and Ca/pectin hydrogel beads loaded with GSE and the spectra of GSE were obtained on an InfraLUM FT-08 FTIR spectrophotometer (Lumex, Saint Petersburg, Russia). The samples were first crushed, mixed with KBr, and pressed. Each spectrum was the result of 20 scans and was acquired at a resolution of 4 cm^−1^ over a range of 4000 to 400 cm^−1^.

### 4.7. Swelling of GSE-Loaded Hydrogel Beads

Wet hydrogel beads were sequentially exposed to artificial gastric (SGF, pH 1.25, for 2 h), intestinal (SIF, pH 7.0, for 4 h), and colonic fluids (SCF, pH 6.8 + pectinase, for 18 h), which were prepared according to [63], with modifications [64]. The wet hydrogel beads (1.2 g, which corresponded to 20 beads) were incubated in simulated fluids (10 mL) at 37 °C and 100 rpm in a shaker incubator (Titramax 1000, Heidolph, Schwabach, Germany) and were collected and weighed at specified time intervals. To determine the swelling ratio (SR), the beads were weighed on an analytical balance after removing excess moisture from their surface. The SR of the beads was calculated using the following equation [48]: SR = W_t_/W_0_, where W_t_ is the weight of the beads after a certain exposure time, and W_0_ is the initial weight.

### 4.8. Determination of Encapsulation Efficiency (EE)

EE was determined using a UV spectrophotometer (SEF-103, Akvilon, Moscow, Russia) at a wavelength of 280 nm in triplicate. EE was determined according to the following formula: EE% = [(Q_t_ − Q_r_)/Q_t_] × 100, where Q_t_ is the amount of initial GSE, and Q_r_ is the sum of the GSE amount recovered in the aqueous solution after filtering and washing the beads.

### 4.9. Cumulative GSE Release from Hydrogel Beads

The hydrogel bead samples were incubated in gastroenteric fluids as described above in Section 4.7. To determine the amount of GSE released from the hydrogel beads, aliquots were removed at specified time intervals, and absorbance was measured at 280 nm in triplicate. All aliquots were then returned to maintain the original volume.

### 4.10. Statistical Analysis

The results obtained are presented as mean ± standard deviation (S.D.). The significance of the differences between the mean values was established using Student’s *t*-test, with significance considered as *p* < 0.05. Correlation coefficients were calculated, and their significance was assessed to determine the relationship between physicochemical characteristics. To examine the rheological characteristics, the significance of differences was assessed using a one-way ANOVA. Statistical calculations were performed using Statistica 10.0 software (StatSoft, Tulsa, OK, USA) and Microsoft Excel 2016.

## Figures and Tables

**Figure 1 gels-10-00273-f001:**
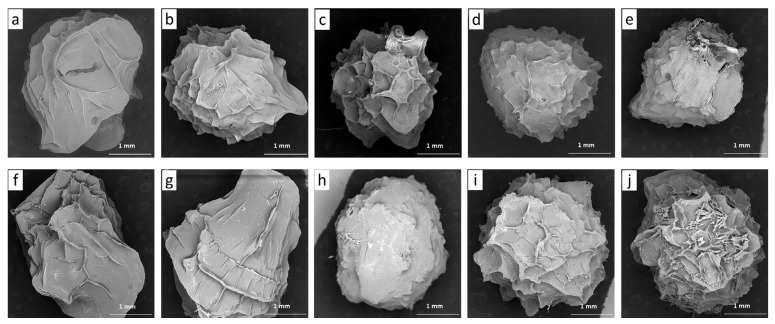
Scanning electron micrographs of GSE-loaded gel beads based on callus cells and TVC (**a**–**e**) or CP (**f**–**j**) pectins: (**a**) 2.0TVC, (**b**) 0.1LM-2.0TVC, (**c**) 0.2LM-2.0TVC, (**d**) 0.3LM-2.0TVC, (**e**) 0.4LM-2.0TVC, (**f**) 2.0CP, (**g**) 0.1LM-2.0CP, (**h**) 0.2LM-2.0CP, (**i**) 0.3LM-2.0CP, and (**j**) 0.4LM-2.0CP. Magnification: 54×; scale bar: 1 mm. Cell-free 2.0TVC and 2.0CP hydrogel beads were used as controls.

**Figure 2 gels-10-00273-f002:**
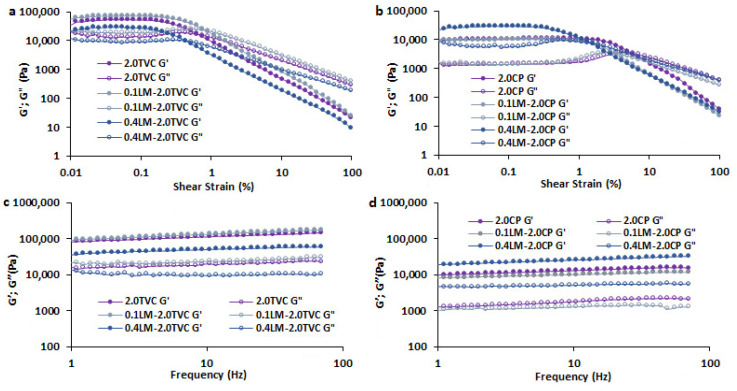
Rheological properties of GSE-loaded pectin hydrogels based on callus cells (0.1 and 0.4 g/mL) and TVC or CP pectins: storage modulus (G′, filled symbols) and loss modulus (G″, empty symbols) test results are represented as a function of shear strain (**a**,**b**) or frequency (**c**,**d**). Cell-free 2.0TVC and 2.0CP hydrogel beads were used as controls.

**Figure 3 gels-10-00273-f003:**
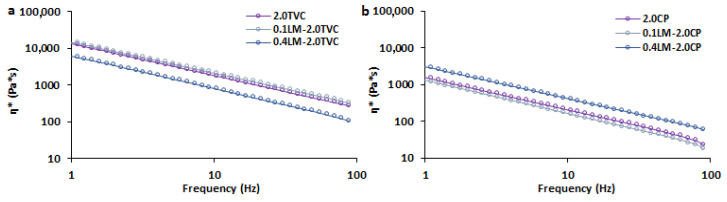
Complex viscosity as a function of frequency for GSE-loaded pectin hydrogels based on callus cells (0.1 and 0.4 g/mL) and TVC (**a**) or CP (**b**) pectins. Cell-free 2.0TVC and 2.0CP hydrogel beads were used as controls.

**Figure 4 gels-10-00273-f004:**
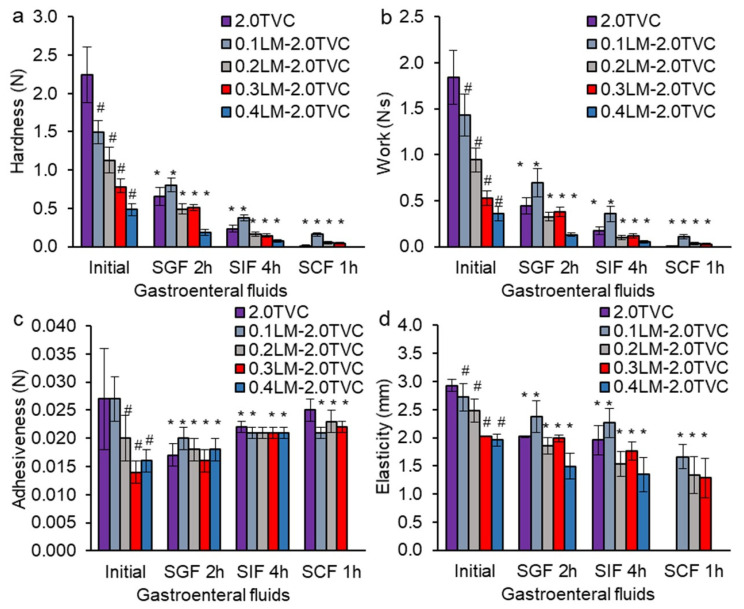
Gel strength (**a**), work (**b**), adhesiveness (**c**), and elasticity (**d**) of GSE-loaded gel beads based on TVC pectin and LM callus cells after successive incubation in SGF, SIF, and SCF. Cell-free 2.0TVC hydrogel beads were used as a control. The data are presented as the mean ± S.D., *n* = 20. # *p* < 0.05 vs. 2.0TVC, * *p* < 0.05 vs. initial characteristics.

**Figure 5 gels-10-00273-f005:**
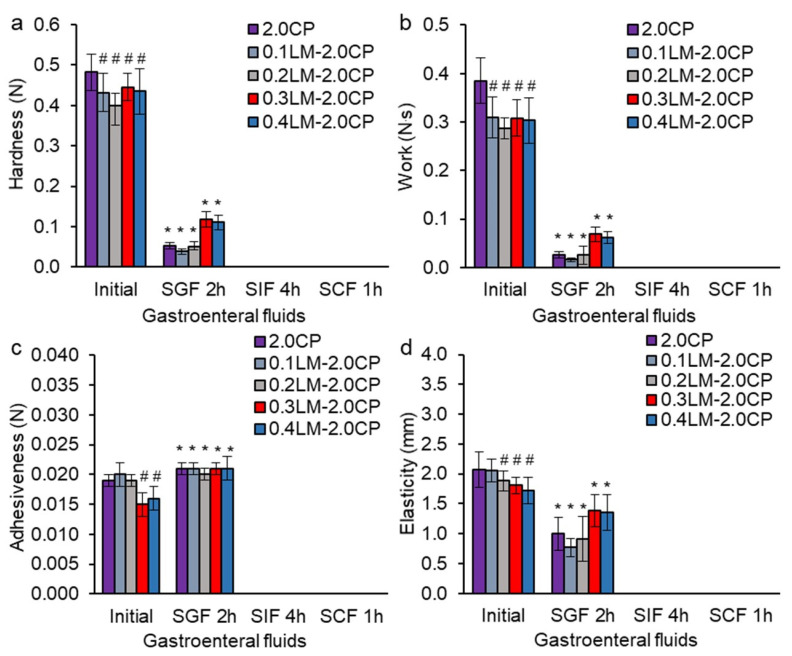
Gel strength (**a**), work (**b**), adhesiveness (**c**), and elasticity (**d**) of GSE-loaded gel beads based on CP pectin and LM callus cells after successive incubation in SGF, SIF, and SCF. Cell-free 2.0CP hydrogel beads were used as a control. The data are presented as the mean ± S.D., *n* = 20. # *p* < 0.05 vs. 2.0CP, * *p* < 0.05 vs. initial characteristics.

**Figure 6 gels-10-00273-f006:**
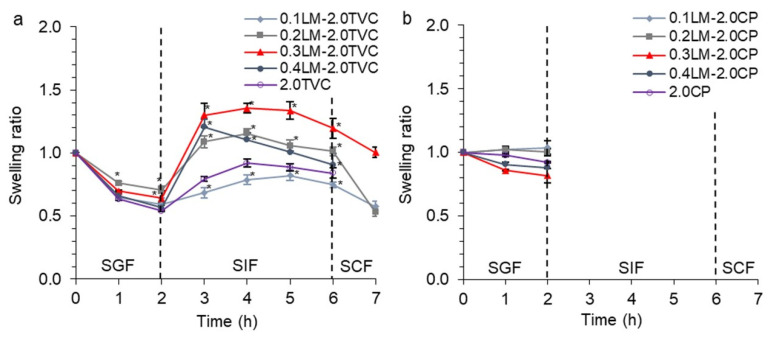
Swelling of GSE-loaded gel beads based on LM callus cells and TVC (**a**) or CP (**b**) pectins in SGF, SIF, and SCF. Cell-free 2.0TVC and 2.0CP hydrogel beads were used as controls. The data are presented as the mean ± S.D., *n* = 15. * *p* < 0.05 vs. 2.0TVC or 2.0CP.

**Figure 7 gels-10-00273-f007:**
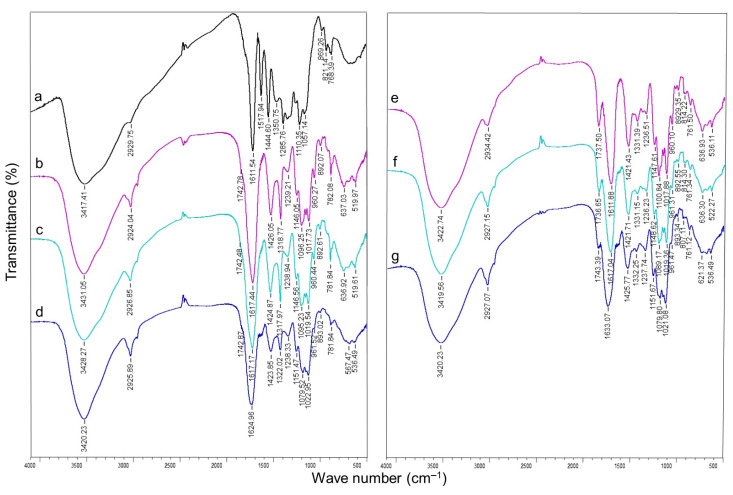
The FTIR spectra of grape seed extract (GSE) (**a**), GSE-loaded cell-free Ca/pectin gel beads (2.0TVC and 2.0CP (controls)) (**b**,**e**), and pectin/callus gel beads (0.1LM-2.0TVC, 0.4LM-2.0TVC, 0.1LM-2.0CP, and 0.4LM-2.0CP) (**c**,**d**,**f**,**g**).

**Figure 8 gels-10-00273-f008:**
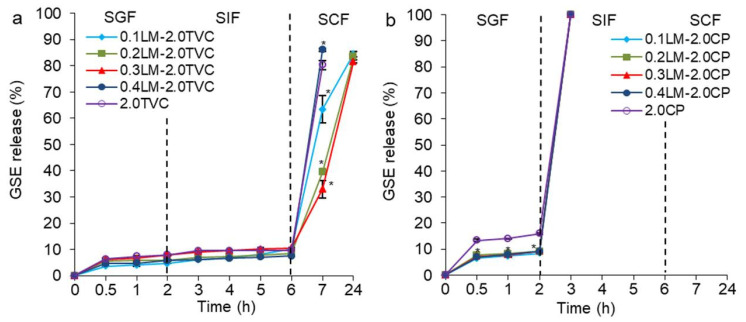
Grape seed extract release from gel beads based on LM callus cells and TVC (**a**) or CP (**b**) pectins in SGF, SIF, and SCF. Cell-free 2.0TVC and 2.0CP hydrogel beads were used as controls. Data are presented as mean ± S.D., *n* = 15. * *p* < 0.05 vs. 2.0TVC or 2.0CP.

**Table 1 gels-10-00273-t001:** Characterization of GSE-loaded hydrogel beads.

Gel Formulation	Content of Callus Cells LM (g/mL)	Concentration of Pectin (%)	Diameter (mm) (*n* = 15)	Phenolic Compounds (μg/mL) (*n* = 4)	Encapsulation Efficiency (%) (*n* = 3)
0.1LM-2.0TVC	0.1	2.0	4.10 ± 0.11 ^a^	40.38 ± 2.91	82.4 ± 0.3 ^a^
0.2LM-2.0TVC	0.2	2.0	4.23 ± 0.10 ^a^	39.43 ± 1.06	79.8 ± 1.1 ^a^
0.3LM-2.0TVC	0.3	2.0	4.42 ± 0.19	43.16 ± 1.61 ^a^	75.9 ± 0.5
0.4LM-2.0TVC	0.4	2.0	4.47 ± 0.20	40.23 ± 2.04	73.2 ± 0.7 ^a^
2.0TVC (control)	0	2.0	4.45 ± 0.16	38.32 ± 1.00 ^b^	75.1 ± 1.2 ^b^
0.1LM-2.0CP	0.1	2.0	4.50 ± 0.11	48.41 ± 1.04 ^b^	73.3 ± 0.3 ^b^
0.2LM-2.0CP	0.2	2.0	4.68 ± 0.12 ^b^	48.22 ± 0.94 ^b^	72.4 ± 0.1 ^b^
0.3LM-2.0CP	0.3	2.0	4.38 ± 0.18 ^b^	51.65 ± 1.28 ^b^	74.5 ± 0.8 ^b^
0.4LM-2.0CP	0.4	2.0	4.43 ± 0.15	51.38 ± 0.43 ^b^	72.1 ± 0.8 ^b^
2.0CP (control)	0	2.0	4.51 ± 0.17	35.74 ± 1.13	57.7 ± 1.1

The data are presented as the mean ± S.D. ^a^ *p* < 0.05 vs. 2.0TVC. ^b^ *p* < 0.05 vs. 2.0CP. LM—cells of *L. minor* callus culture. Cell-free 2.0TVC and 2.0CP hydrogel beads were used as controls.

**Table 2 gels-10-00273-t002:** The rheological characteristics of GSE-loaded Ca/pectin (2.0TVC and 2.0CP) and pectin/callus hydrogels (0.1LM-2.0TVC, 0.4LM-2.0TVC, 0.1LM-2.0CP, and 0.4LM-2.0CP).

Parameters	2.0TVC (Control)	0.1LM-2.0TVC	0.4LM-2.0TVC	2.0CP (Control)	0.1LM-2.0CP	0.4LM-2.0CP
Strength of linkage						
G′_LVE_ (Pa)	52,066 ± 3107 ^a^	78,174 ± 3549 ^b^	26,892 ± 2414 ^c^	10,443 ± 362 ^a#^	9462 ± 313 ^b^*	31,660 ± 1294 ^c@^
G*_FP_ (Pa)	54,239 ± 14,714 ^a^	70,329 ± 20,984 ^a^	28,147 ± 10,688 ^b^	10,545 ± 360 ^a#^	9727 ± 1358 ^a^*	28,092 ± 4322 ^b^
Tan [δ]_AF_	0.21 ± 0.15 ^a^	0.15 ± 0.06 ^a^	0.20 ± 0.06 ^a^	0.10 ± 0.02 ^a#^	0.13 ± 0.04 ^a^	0.30 ± 0.33 ^a^
τL (Pa)	47,129 ± 10,915 ^a^	57,033 ± 27,952 ^a^	21,097 ± 6958 ^b^	10,449 ± 820 ^a#^	9228 ± 421 ^a^*	29,716 ± 4297 ^b^
k′ (Pa·s)	79,941 ± 21,810 ^a^	87,802 ± 4942 ^a^	35,483 ± 5253 ^b^	9817 ± 1466 ^a#^	7926 ± 734 ^a^*	18,415 ± 8236 ^b@^
k″(Pa·s)	16,037 ± 7519 ^ab^	20,955 ± 4197 ^a^	11,440 ± 4967 ^b^	1286 ± 217 ^a#^	1171 ± 184 ^a^*	4653 ± 1177 ^b@^
A (Pa·s)	87,807 ± 18,799 ^a^	91,010 ± 3803 ^a^	37,741 ± 6349 ^b^	9859 ± 1493 ^a#^	8070 ± 868 ^a^*	19,132 ± 8262 ^@b^
Number of linkages						
G*max/G*_LVE_	1.07 ± 0.03 ^a^	1.09 ± 0.06 ^a^	1.10 ± 0.07 ^a^	1.05 ± 0.01 ^a^	1.04 ± 0.01 ^a^	1.03 ± 0.02 ^a^
τFr (Pa)	16,208 ± 2833 ^ab^	21,872 ± 9699 ^a^	9264 ± 2699 ^b^	3317 ± 528 ^a#^	3582 ± 270 ^a^*	9335 ± 2484 ^b^
n′	0.14 ± 0.02 ^a^	0.18 ± 0.03 ^a^	0.14 ± 0.03 ^a^	0.12 ± 0.01 ^ab^	0.10 ± 0.01 ^a^*	0.14 ± 0.03 ^b^
z	7.11 ± 1.34 ^ab^	6.06 ± 0.83 ^a^	8.08 ± 1.42 ^b^	8.17 ± 0.52 ^ab^	10.30 ± 1.75 ^a^*	7.72 ± 1.45 ^b^
Timescale of junction zone						
Tan [δ]_LVE_	0.28 ± 0.05 ^a^	0.26 ± 0.03 ^a^	0.35 ± 0.07 ^b^	0.10 ± 0.02 ^a#^	0.15 ± 0.01 ^a^*	0.21 ± 0.03 ^b@^
k″/k′	0.19 ± 0.04 ^a^	0.24 ± 0.06 ^ab^	0.32 ± 0.11 ^b^	0.13 ± 0.01 ^a#^	0.15 ± 0.01 ^a^*	0.28 ± 0.11 ^b@^
η*s (Pa·s)	12,099 ± 2565 ^a^	14,184 ± 673 ^a^	6030 ± 1035 ^b^	1569 ± 238 ^a#^	1020 ± 492 ^a^*	3045 ± 1315 ^b@^
γL (%)	0.20 ± 0.06 ^a^	0.21 ± 0.05 ^a^	0.19 ± 0.05 ^a^	1.33 ± 0.63 ^a#^	1.08 ± 0.33 ^a^*	0.33 ± 0.29 ^b^
γFr (%)	0.65 ± 0.19 ^a^	0.77 ± 0.27 ^a^	0.42 ± 0.10 ^a^	5.62 ± 1.63 ^a#^	3.03 ± 0.44 ^b^*	1.16 ± 0.59 ^c@^
Distance of linkage						
n″	0.07 ± 0.03 ^a^	0.06 ± 0.04 ^a^	0.06 ± 0.04 ^a^	0.96 ± 0.02 ^a#^	0.35 ± 0.22 ^b^	0.03 ± 0.03 ^b^

The data are presented as the mean ± S.D., *n* = 5. Significant differences (*p* < 0.05) between means are labeled with different lowercase letters (a, b, and c). ^#^ *p* < 0.05 vs. 2.0TVC, * *p* < 0.05 vs. 0.1LM-2.0TVC, ^@^ *p* < 0.05 vs. 0.4LM-2.0TVC. The storage modulus (G′_LVE_), corresponding complex modulus (G*_FP_) with the stress at flow point, the slope of the loss tangent after flow point (Tan [δ]_AF_), limiting value of stress (τL), G*max/G*_LVE_ (the ratio of maximum complex modulus to linear complex modulus), the fracture stress (τFr), the loss tangent (Tan [δ]_LVE_), limiting value of strain (γL), and the fracture strain (γFr) are shown. The frequency dependences of the elastic (k′ and n′), loss (k″ and n″), and complex (A and z) moduli; overall loss tangent (k″/k′); and the slope of complex viscosity (η*s) are also shown. The pectin/callus hydrogels were based on callus cells (0.1 and 0.4 g/mL) and TVC or CP pectins. Cell-free 2.0TVC and 2.0CP hydrogel beads were used as controls.

**Table 3 gels-10-00273-t003:** Characterization of pectins.

Pectin	Contents (% of Total Amount)								Mw (kDa)	DM (%)
	GalA	Gal	Ara	Rha	Glc	Xyl	Man	Protein		
TVC	89.2	2.2	1.6	0.9	0.9	0.4	0.1	4.8	666	5.7
CP	71.0	1.5	0.3	1.0	24.3	0.8	0	1.2	402	33.0

GalA—galacturonic acid; Gal—galactose; Ara—arabinose; Rha—rhamnose; Glc—glucose; Xyl—xylose; Man—mannose; Mw—molecular weight; DM—degree of methylesterification.

## Data Availability

The data that support the findings of this manuscript are available from the corresponding author upon reasonable request.

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
