# Peer review of "Effect of Callus Cell Immobilization on the Textural and Rheological Properties, Loading, and Releasing of Grape Seed Extract from Pectin Hydrogels"

_gels, 2024, doi:10.3390/gels10040273_

Round 1
Reviewer 1 Report
Comments and Suggestions for Authors
overall article is well organized and recommended for publication after a minor revision. The specific comments are given below:
1. In Table 1 use of statistical lettering should be evaluated again
2. Table 2, the units should be added to know the measurements
3. All tables footnotes could be improved, based on the abbreviations used in tables
Figure 7. It would be nice if authors could redraw FTIR spectra by overlaying all samples like other figures, which could be more comparable among other samples
Line 311: Replace SGF with SCF please as SGF is repeated in this sentence
Regards,
Good Luck
Reviewer 2 Report
Comments and Suggestions for Authors
Dear Authors; Re: [Manuscript ID gels-2963779]
Title: "Effect of callus cell immobilization on the textural and rheological properties, loading and releasing of grape seed extract from pectin hydrogels"
In your original research article you aimed to formulate and prepare Hydrogel beads based on pectins of different structures and immobilized callus cells and characterise them in vitro.
Please note the following:
1. Rationale behind immobilizing Callus Cells not properly described.
2. Please double check 1st sentence of Abstract and last sentence of Introduction and rephrase them in a way to be clear and avoiding confusion for readers.
3. Stability evaluations not mentioned.
4. Controls are not shown in the results.
5. You have performed size analysis, however Particle size data not shown.
6. Add Particle number per volume using Simple Equations Pertaining to the Particle Number and Surface Area ...
7. In Figure 7 (FTIR Spectra) the characters are too small.
8. A List of Abbreviations will benefit readers.
9. Try to add more recent References (from 2024 only one paper is cited).
Thank you
